# Effect of Preparation Conditions on Application Properties of Environment Friendly Polymer Soil Consolidation Agent

**DOI:** 10.3390/polym14102122

**Published:** 2022-05-23

**Authors:** Shaoli Wang, Shengju Song, Xuping Yang, Zhengqi Xiong, Chaoxing Luo, Yongxiu Xia, Donglu Wei, Shaobo Wang, Lili Liu, Hong Wang, Lifang Sun, Lichao Du, Shaofeng Li

**Affiliations:** 1State Key Laboratory of Tree Genetics and Breeding, Experimental Center of Forestry in North China, National Permanent Scientific Research Base for Warm Temperate Zone Forestry of Jiulong Mountain in Beijing, Chinese Academy of Forestry, Beijing 100091, China; wshaoli@iccas.ac.cn (S.W.); yxxia@caf.ac.cn (Y.X.); slf2014@163.com (L.S.); 2R & D Department, China Academy of Launch Vehicle Technology, Beijing 100076, China; songshengju99@163.com (S.S.); duchao0620@163.com (L.D.); 3Chinese Academy of Forestry, Beijing 100091, China; yxp@caf.ac.cn; 4College of Material and Chemical Engineering, Heilongjiang Institute of Technology, Harbin 150050, China; xiongzhengqi1020@163.com (Z.X.); luochaoxing525@163.com (C.L.); mit_ing@163.com (D.W.); liulili760802@126.com (L.L.); wanghongg357@126.com (H.W.); 5Beijing Yangsheng New Material Technology Co., Ltd., Beijing 102299, China; wshaobo2009@163.com

**Keywords:** polymer soil consolidation agent, preparation conditions, consolidated adhesive film, compressive strength, seedling transplanting

## Abstract

In order to improve the survival rate of transplanted seedlings and improve the efficiency of seedling transplantation, we developed an environmental friendly polymer konjac glucomannan (KGM)/chitosan (CA)/polyvinyl alcohol (PVA) ternary blend soil consolidation agent to consolidate the soil ball at the root of transplanted seedlings. In the previous research, we found that although the prepared KGM/CA/PVA ternary blend soil consolidation agent can consolidate the soil ball at the root of the seedling, the medium solid content of the adhesive was high, which affects its spraying at the root of the seedling. At the same time, the preparation temperature of the KGM/CA/PVA ternary blend was also high. Therefore, to reduce the energy consumption and the cost of the KGM/CA/PVA ternary blend soil consolidation agent in the preparation process, this paper studied the influence of preparation conditions on the application performance of the environmental friendly polymer soil consolidation agent. We aimed to reduce the highest value CA content and preparation temperature of the KGM/CA/PVA ternary blend adhesive on the premise of ensuring the consolidation performance of the KGM/CA/PVA ternary blend adhesive on soil balls. It was prepared for the popularization and application of the environmental friendly polymer KGM/CA/PVA ternary blend soil consolidation agent in seedling transplanting. Through this study, it was found that the film-forming performance of the adhesive was better when the KGM content was 4.5%, the CA content was in the range of 2–3%, the PVA content was in the range of 3–4%, and the preparation temperature was higher than 50 °C. The polymer soil consolidation agent prepared under this condition has a good application prospect in seedling transplanting.

## 1. Introduction

Both barren mountain afforestation and urban greening involve seedling transplanting [1]. In order to improve the survival rate of transplanted seedlings, the most important thing is to ensure the integrity of the soil ball at the root of transplanted seedlings. The diameter of the soil ball at the root of the seedling is generally required to be 5–10 times the diameter at breast height (DBH) of the tree itself, and the height of the soil ball is generally about 2/3 of its diameter. After successful lifting from the nursery, the soil ball is very easy to break due to squeezing, bumping, and mutual impact during its transportation. To ensure the integrity of transplanted seedlings, the commonly used methods are wrapping, binding, or wooden box packaging [1,2]. There are two problems in the current method to ensure the integrity of the soil ball. First, some materials wrapped and bound, such as plastic rope, plastic cloth, and iron wire, do not degrade easily and can cause environmental pollution. Second, the technical requirements for soil ball wrapping, binding, and seedling raising are high, so it not only consumes a lot of manpower and material resources but also has low seedling raising efficiency. The key is that it is difficult to ensure the integrity of the soil ball in the end, especially in the process of loading, unloading, and transporting seedlings where the mother soil ball is often damaged.

In view of the demand for seedling transplanting and maintaining the integrity of the mother soil ball, our research team prepared KGM/CA binary blend soil consolidation agents and KGM/CA/PVA ternary blend soil consolidation agents with KGM, CA, and PVA as the main raw materials in the previous research work [3,4,5]. The soil consolidation agent was evenly sprayed on the surface of the soil ball at the root of the transplanted seedlings. After the consolidation agent was dry, it could form a hard film on the surface of the soil ball. In the previous research, it was found that the film formed on the surface of the soil ball by KGM/CA and KGM/CA/PVA blend soil consolidation agents can protect the soil ball in the process of transportation, which reduced wear and improved its resistance to transportation oscillation [3,4,5]. The application of the KGM/CA and KGM/CA/PVA blend consolidation agent also had no adverse effect on the growth of the seedlings.

KGM, the raw material for preparing the soil consolidation agent, is a rich natural macromolecular polysaccharide, which can be extracted from amorphophallus konjac plant tubers [6,7,8,9,10]. KGM is mainly composed of D-mannose and D-glucose units through β-1,4-linkages in a mole ratio of 1.6:1 [11,12,13]. Due to the presence of active primary hydroxyl (*-CH_2_OH*) at the *C*(6) position of each sugar unit, KGM can participate in many chemical reactions, such as nitration, etherification, and graft polymerization [14]. The *O*-acetyl (*-OCH_3_*) group located at the *C*(6) position of the sugar residues contributes to the solubility and swelling of KGM [15,16]. KGM is water-soluble and has high viscosity even at low concentrations. It can also form a gel network structure through extensive hydrogen bonding and entanglement [17]. Therefore, KGM is very suitable for preparing soil consolidation agents. However, the glue solution prepared by a single KGM can not meet the consolidation of soil. Considering the convenience and economy of preparing soil consolidation agents, it is wise to use physical blending modification technology to improve the consolidation performance of soil.

There are two directions in the physical blending modification of KGM. One is to blend with other gel polysaccharides, chitosan, starch, and other substances, so as to improve the viscosity or gel strength of products [18,19,20,21]. Second, KGM is blended with other synthetic polymers to give it new functions. In the process of blending, the addition of other polymer materials can greatly improve the hydrogen bonding in KGM molecules and form a new spatial network structure [22,23].

PVA contains a large number of hydroxyl groups in its molecular chain, which can be crosslinked to form a macromolecular network. Meanwhile, PVA is a water-soluble polymer material with good biocompatibility and film-forming properties [24,25]. After blending with PVA, the mechanical properties of the KGM film can be improved.

CA is the product of deacetylation of chitin macromolecule, and its structure is similar to cellulose [26,27]. Many hydroxyl and amino groups are distributed on the macromolecular chain of chitosan, which gives it good solubility and reactivity. CA also has good biocompatibility, adsorption, film-forming, permeability, moisture retention, and biodegradability properties [27,28]. Blending CA with KGM and PVA can improve the adhesion of the glue solution on the soil surface, film-forming, moisturizing, and air permeability of the glue film. However, because the price of CA is higher than that of KGM and PVA, it is necessary to explore the influence of the solid content of CA in the KGM/CA/PVA ternary blend on the application performance of the soil consolidation agent, which is of great significance for the market application of KGM/CA/PVA ternary blend soil consolidation agents in the future.

In the preparation process of the soil consolidation agent, it was found that the preparation temperature of the adhesive had an obvious influence on the viscosity and film-forming of the KGM/CA/PVA ternary blend adhesive, as well as the consolidation of soil and the anti-compression and anti-transport oscillation of the consolidated soil column. The higher the preparation temperature of the KGM/CA/PVA ternary blend adhesive, the higher the preparation cost and an increase in energy consumption, which is not conducive to reducing carbon emission. At the same time, the content of CA and PVA can not only affect the viscosity, fluidity, film-forming, and consolidation of the soil ball but also affect the preparation cost of the KGM/CA/PVA consolidation agent.

Therefore, in order to reduce the energy consumption and cost of the KGM/CA/PVA ternary blend soil consolidation agent in the preparation process, this paper studied the influence of its preparation conditions, such as preparation temperature, CA and PVA content, on the application performance of environmental friendly polymer KGM/CA/PVA soil consolidation agent. It can also reduce the content of CA in the adhesive and the preparation temperature of polymer soil consolidation agent on the premise of ensuring the consolidation performance of KGM/CA/PVA ternary blend soil consolidation agent on soil balls. It is prepared for the popularization and application of the environmental friendly polymer KGM/CA/PVA ternary blend soil consolidation agent in seedling transplanting.

## 2. Materials and Methods

### 2.1. Materials and Experimental Instruments

(1)Materials: The main raw material konjac flour (KGM, 200 g/bottle) was provided by Bozhou Baofeng bio-technology limited company. Chitosan (CA, chemical pure), polyvinyl alcohol (PVA, superior-grade pure), acetic acid (excellent-grade pure), sodium hydroxide (analytical purity), and other compounds were supplied by Sinopharm Chemical Reagent Co., Ltd. of China, Shanghai, China.(2)Experimental instruments: Water bath heating pot; Mechanical agitator; Automatic film-coating machine; Stereomicroscope (Leica DFC425C); Mechanical testing machine INSTRON 5582; Simulated transportation vibration test bench hk-120 with a payload of 300 kg.

### 2.2. Preparation and Viscosity Test Method of Environmental Friendly Soil Consolidation Agent

(1)Preparation of polyvinyl alcohol (PVA) solution: Add 360 g of ultrapure water to the neck mouth flask, add 40 g of PVA (molecular weight: 1840 g/mol) to the four mouth flask, and then put the flask into a water bath, heat it to the temperature of 95 °C, and stir mechanically (rotating speed 150 r/min) for 2 h to obtain a PVA solution with a mass fraction of 10%.(2)Preparation of chitosan solution: Weigh a certain amount of CA into a four neck flask, add a pre-configured acetic acid solution with a mass fraction of 20%, stir until it is evenly dissolved to obtain a dilute acid solution of CA.(3)Preparation of KGM/CA/PVA ternary blending soil consolidation agent: Weigh a certain amount of KGM into a four neck flask, add an appropriate amount of CA dilute acid solution according to the preset proportion, and mechanically stir (400 r/min) for a certain time at the target temperature until KGM and CA in the system are completely dissolved. Then the NaOH was added to the system to adjust the pH value of the reaction solution to 4.2–4.5. Finally, KGM/CA/PVA ternary blend adhesive was obtained by adding 10% polyvinyl alcohol solution prepared in advance and stirring for 1 h. Then, according to the actual needs, tackifier and preservative can be added to KGM/CA/PVA ternary blend adhesive. Finally, transfer the KGM/CA/PVA ternary blend adhesive to a wide mouth bottle and seal it for standby after it is reduced to room temperature.(4)Test method for viscosity of KGM/CA/PVA ternary blend adhesive: Using Brooke digital display DV3 viscometer, immerse the rotor into the ternary blend adhesive of different formulations at room temperature, select rotor No. 64, set the speed to 10 r/min to obtain the viscosity of the ternary blend adhesive.

### 2.3. Preparation and Test of KGM/CA/PVA Ternary Blend Film

(1)Preparation of KGM/CA/PVA ternary blend film: Weigh 5 g KGM/CA/PVA ternary blend adhesive, and then pour the adhesive directly onto the substrate in the middle of the automatic coating machine console. Select the appropriate scraper and the appropriate mode. The scraper will move back and forth on the track at a certain speed until the KGM/CA/PVA ternary blend adhesive is evenly coated on the substrate. Then, the substrate with the adhesive film is translated to a dry position and naturally dried to form a film.(2)The SEM image of KGM/CA/PVA ternary blend film: After the prepared KGM/CA/PVA ternary blend film was dried in a dryer for 24 h, the film with uniform appearance was selected and cut into small strips, and the sample for observing the upper surface was prepared.In addition, the membrane at the same part were brittle broken after liquid nitrogen freezing, and the cross-section port was scanned. Place the pasted sample on the copper table, spray gold under 13.3 Pa vacuum for 20–30 s, and the thickness is about 690 nm. Under the condition of accelerating voltage of 20 kV, the surface and cross-section morphology of the film samples were observed by scanning electron microscope (SEM).

### 2.4. Preparation and Tests of Soil Column Samples

(1)Preparation of soil column samples: Two different types of consolidated soil columns were prepared from cinnamon soil (loam) with different formulations of KGM/CA/PVA ternary blend adhesive. The preparation method is similar to the original [3].(2)Optical characterization of KGM/CA/PVA ternary blend adhesive film on the surface of consolidated soil column: Referring to the original characterization method [3,5], Leica DFC425C stereoscope was used to observe the surface morphology of the bonding film on the upper surface of the consolidated soil column. Obtain the best image of the consolidated adhesive film on the upper surface of the soil column sample by adjusting the “light source” and “focusing/zoom”, and transmit the image to the picture window of the software for saving.(3)Test method of compressive strength of consolidated soil column: Referring to the original test method [3,5], the compressive strength of consolidated soil column is tested by INSTRON 5582 universal testing machine. The test conditions are that the time interval is 0.5 s and the compression rate is 1 mm/min.(4)Test method of anti-transport oscillation of consolidated soil column: Referring to the original test method [3], according to the American Transportation Association standard (ISTA) and American Society of materials standard (ASTM), the transportation oscillation resistance of consolidated soil column is tested by simulated transportation vibration test bench HK-120, as shown in Table 1.

### 2.5. Method of Transplanting Seedlings with KGM/CA/PVA Ternary Blend Adhesive in Seedling Transplanting

The application of ternary blend soil consolidation agent for seedling transplanting adopts a relatively simple method: Take loam as the main consolidation object and seedlings with DBH of 5–10 mm growing outdoors as the main transplanting object. Firstly, circle a circle with a diameter of about 10–15 cm with the transplanted seedling as the center at the root of the seedling, and then gradually clean the soil outside the circle from top to bottom until a conical soil ball is formed at the root of the transplanted seedling. There is about 1 cm contact surface between the bottom of the conical soil ball and the earth. Then, the soil consolidation agent is sprayed on the upper and side surfaces of the conical soil ball. After about 1 day of consolidation, the glue liquid on the surface of the soil ball will form a hard shell on the surface of the soil ball together with the soil on the surface of the soil ball. After the soil ball is lifted, it can be directly transported to a new planting site for transplanting. When transplanting seedlings in sandy soil with the soil consolidation agent, except that the method of preparing the soil ball is slightly different, the transplanting steps are similar to those in loam [3,4,5].

## 3. Results

### 3.1. Effect of Preparation Conditions on Viscosity of KGM/CA/PVA Ternary Blend Adhesive

In previous studies, it was found that the preparation temperature and solid content of the blend adhesive had a great influence on the viscosity and fluidity of the blend adhesive [3,4,5]. In order to reduce the content of CA in the ternary blend solution and ensure the viscosity of the solution to the consolidation of soil columns, the effect of CA content on the viscosity of the solution at different temperatures was explored.

It can be seen from Figure 1a,b that the viscosity of the KGM/CA/PVA blend adhesive increases gradually with the increase of CA content in the glue at different temperatures. It was found that the viscosity increased slowly when the CA content was less than 1.5% at the preparation temperature of 40 °C; when the CA content was higher than 1.5%, the viscosity of the adhesive increased rapidly, as shown in Figure 1a. When the preparation temperature was 50 °C, and the CA content of the KGM/CA/PVA blend adhesive was lower than 3%, the viscosity increased rapidly; when the CA content was higher than 3%, the increase of adhesive viscosity slowed down, as shown in Figure 1b. Therefore, we explored the effect of CA content (2–3%) in the glue solution on the viscosity of the blend adhesive system at different temperatures, as shown in Figure 1c. It was found that when the CA content was fixed, with the increase of the preparation temperature, the viscosity of the adhesive first increased and then decreased, and the viscosity was highest at 60 °C. When the CA content in the blend adhesive was 3%, the viscosity in the blend adhesive system was highest as a whole. When the preparation temperature of the glue was at 60 °C, the viscosity of the adhesive reached a maximum value. It can be seen that the preparation temperature and the CA content in the adhesive have a great influence on the viscosity of the KGM/CA/PVA blend adhesive. Excessive viscosity of the KGM/CA/PVA blend adhesive can affect its fluidity, so it is necessary to select the appropriate preparation conditions according to the consolidation demand of the soil.

### 3.2. Effect of Preparation Conditions on Properties of KGM/CA/PVA Ternary Blend Film

The preparation conditions of the KGM/CA/PVA ternary blend adhesive, such as preparation temperature, CA content, and PVA content, have a great impact not only on the viscosity but also on the film-forming property, strength, toughness, and internal structure of the film.

#### 3.2.1. Effect of Preparation Conditions of KGM/CA/PVA Ternary Blend Adhesive on Film-Forming Properties

The prepared ternary blend adhesive was prepared into a thin film by an automatic coating machine. When the KGM/CA/PVA ternary blend adhesive with different CA content was prepared into a glue film at 40 °C, it was found that when the CA content was 0%, the formed thin film was relatively brittle and it was difficult to form a whole film, as shown in Figure 2a. From Figure 2b,c, it was found that the brittleness of the film was gradually improved, and the formed film was gradually complete with the gradual increase of CA content. However, due to the low preparation temperature of the KGM/CA/PVA ternary blend adhesive, the quality of the formed adhesive film was relatively poor. With the increase in temperature, the viscosity of the KGM/CA/PVA ternary blend adhesive increases, and the toughness of the film increased to a certain extent. As shown in Figure 2d, the adhesive film formed by the adhesive prepared under the condition of 50 °C had a surface of well-distributed pore, which met the application requirements of the consolidated soil ball. At the same time, the toughness of the adhesive film was good, and the complete membrane can be cut evenly with a die. As the temperature continued to rise to 60 °C, the formed adhesive film was more flexible and better, as shown in Figure 2e. When the temperature reached to 70 °C, the formed adhesive film can be bent at multiple angles, as shown in Figure 2f.

#### 3.2.2. Effect of Preparation Conditions of KGM/CA/PVA Ternary Blend Adhesive on the Structure of Film

The effect of temperature on the internal structure of the film can be seen from the surface and cross-section of the film. The adhesive film formed by the KGM/CA/PVA ternary blend adhesive was prepared at 40 °C. As shown in Figure 3a, it can be seen that the adhesive film was not densely arranged, was unevenly distributed, and had many cavities. As shown in Figure 3b, it can be seen that the links between the cured products on the surface of the adhesive film were weak, the pores formed were large, and there was no cross-linking between the molecular chains, so the adhesive film formed was brittle. From Figure 3c,d, it can be seen that when the temperature for preparing the ternary blend adhesive was increased to 70 °C, the surface of the adhesive film was densely arranged, and there were many short protrusions on the surface and connected into a network structure. At the same time, there were a small number of pores on the surface of the formed adhesive film, which can meet the application requirements of the transplanted seedlings consolidated soil ball. Through Figure 3e,f, it can be seen that KGM, CA, and PVA were cross-linked together, permeated, and interspersed with each other, which reflected the enhanced interaction between KGM, CA, and PVA. It can be seen from Figure 3f that an interpenetrating structure was formed between KGM, CA, and PVA, indicating that KGM, CA, and PVA have good compatibility. This morphology not only ensured the bonding strength but also improved the water resistance to a certain extent.

### 3.3. Effect of Preparation Conditions on the Morphology of Consolidated Adhesive Film

The loam in cinnamon soil was prepared into the soil column by mold, and KGM/CA/PVA ternary blend adhesive was evenly sprayed on the surface of the soil column to form a uniform glue film. The soil column at this stage contains a certain amount of water. During the drying of the soil column, with the volatilization of water, the morphology of the polymer consolidated film covering the surface of the soil column can change, and produce bubbles similar to fish eye. The moisture permeability of the outer polymer film can be directly judged by observing the morphology of the solid film on the surface of the soil column through the body microscope. It was found that the preparation temperature and the content of CA in the KGM/CA/PVA ternary blend can affect the surface morphology of the consolidated film. When the preparation temperature was 50 °C, relatively large holes were easier to form on the surface of the consolidated adhesive film, as shown in Figure 4a–d. When the CA content was low, a reticular membrane with large pores was formed on the surface of the soil column, as shown in Figure 4a. As the CA content gradually increased from 1% to 3%, the holes on the surface of the consolidated adhesive film gradually shrank and formed many small bubbles, as shown in Figure 4b–d. When the preparation temperature of the glue solution rose to 60 °C and 70 °C, it was found that with the increase of CA content, the bubbles on the surface of the consolidated adhesive film were also gradually decreasing, as shown in Figure 4e–l. It can be seen that in the KGM/CA/PVA ternary blend adhesive, with the increase of CA content, the film-forming performance of the soil column consolidated adhesive film was better. Therefore, under the same humidity conditions, its bubbles gradually decreased.

It was found that when the CA content in the KGM/CA/PVA ternary blend adhesive was the same, the bubbles on the surface of the consolidated film also showed a gradually decreasing trend with the preparation temperature of the solution rising from 50 °C to 70 °C, as shown in Figure 4a,e,i, and Figure 4b,f,j. This was because, with the increase in temperature, CA dissolved more thoroughly in the blend adhesive. CA formed a short cross-linking network structure after blending with KGM and PVA. The film-forming performance of the KGM/CA/PVA ternary blend was better. Therefore, under the same humidity conditions, the bubbles on the surface of the consolidated film were small.

### 3.4. Effect of Preparation Conditions on Compressive Properties of Consolidated Soil Columns

It was found that the preparation temperature and solid content of the KGM/CA/PVA ternary blend adhesive had great effects on the compressive properties of the consolidated soil columns. The film-forming performance of PVA is good, but when the concentration of PVA is large, the viscosity is too large, which is not conducive to the fluidity of glue solution. Therefore, it is necessary to study the influence of PVA content on the compression resistance of consolidated soil columns. It was found that the compressive strength of consolidated soil columns increased with the increase of PVA content from 1% to 5% at the same temperature, as shown in Figure 5a. At the same time, it was also found that when the PVA content was fixed, the compressive strength of the consolidated soil column increased with the increase of the preparation temperature from 40 °C to 70 °C, especially when the temperature was higher than 50 °C, the compressive strength of the consolidated soil column increased rapidly, and the maximum compressive strength reached 4.57 MPa. It can be seen that the preparation temperature and PVA content of the KGM/CA/PVA ternary blend adhesive can directly affect the compressive strength of the consolidated soil column.

Due to the high price of CA, in order to reduce the preparation cost of KGM/CA/PVA ternary blend soil consolidation agent, it is necessary to study the effect of the reduction of CA content in the glue on the compressive properties of consolidated soil columns. It was found that at the same temperature, with the decrease of CA content, the compressive strength of the consolidated soil column also showed a decreasing trend, as shown in Figure 5b. However, at 60 °C, with the decrease of CA content, the compressive strength of the consolidated soil column decreased relatively little. At the same time, it was also found that the compressive strength of the consolidated soil column was higher with the increase in the preparation temperature of the glue solution. Therefore, it is necessary to select the appropriate preparation temperature and CA content according to the actual application requirements.

### 3.5. Effect of Preparation Conditions on Anti-Transport Oscillation of Consolidated Soil Column

During the transportation of transplanted seedlings, the consolidated soil ball needs to bear various oscillations during the transportation from the nursery to the planting point. According to the effect of the preparation conditions of the KGM/CA/PVA ternary blend adhesive on the viscosity of glue and the anti-compression performance of the consolidated soil column, we selected three better conditions for preparing KGM/CA/PVA adhesive, as shown in Table 2. In order to study the influence of different preparation conditions on the anti-transportation oscillation of the consolidated soil column, we used the simulated transportation vibration test-bed HK-120 to test the anti-transportation oscillation of the consolidated soil column. The oscillation conditions were shown in Table 1. As shown in Figure 6, in oscillation experiment 1 (180 rpm, 3.0 Hz, 79 min), it can be seen that samples a_2_, b_2_, c_2_ and d_2_ have almost no wear; In oscillation experiment 2 (210 rpm, 3.5 Hz, 66 min), compared with the control sample a_3_, it can be found that the wear of sample b_3_ was very small; The wear of samples c_3_ and d_3_ were larger than that of b_3_ but smaller than that of sample a_3_; In oscillation test 3 (240 rpm, 4.0 Hz, 60 min), with the continuous increase of amplitude, the surface of the four soil column samples showed obvious wear. However, it can be clearly seen that the wear degree of soil column samples b_4_, c_4,_ and d_4_ consolidated with glue was less than that of the control sample a_4_. It can be seen that the KGM/CA/PVA ternary blend film formed on the surface of the consolidated soil column has a good protective effect on the soil column in the process of transportation oscillation.

### 3.6. Preliminary Application of Polymer Soil Consolidation Agent in Seedling Transplanting

In order to verify that the KGM/CA/PVA ternary blend adhesive can indeed consolidate the loam soil ball at the root of seedlings, we conducted a preliminary application study on the prepared polymer soil consolidation agent in the actual transplanting process of seedlings. Firstly, during the transplanting process of sierra salvia and Euonymus japonicas, we sprayed the KGM/CA/PVA ternary blend adhesive on the soil ball at the root of sierra salvia or Euonymus japonicas. After the blend adhesive dried, a layer of consolidated film formed on the surface of the soil ball. At this time, the plant sierra salvia or Euonymus japonicas can be transplanted directly with the soil ball, as shown in Figure 7a,b. It was found that the blend adhesive can consolidate the soil ball with a diameter of about 20 cm. After observation, the consolidated glue had no adverse effect on the growth of sierra salvia and Euonymus japonicas. It can be concluded that the KGM/CA/PVA ternary blend adhesive prepared by us has a good consolidation effect on loam and brings convenience to the process of seedling transplantation.

In order to further verify the consolidation effect of the KGM/CA/PVA ternary blend adhesive on sandy soil, we sprayed the adhesive on the surface of the sandy soil ball during the transplanting of Juniperus sabina. After drying, a layer of consolidation glue film was formed on the surface of the sandy soil ball, as shown in Figure 7c. At this time, you can directly carry the Juniperus Sabina seedlings and soil balls to transplant Juniperus Sabina to the planting site. To further verify the consolidation effect of the glue on sandy soil, we sprayed the glue on the sandy soil ball with a diameter of about 15–30 cm at the root of Euonymus japonicus. After drying, the glue formed a relatively hard transparent polymer glue film on the surface of the sandy soil ball. Due to the protective effect of the adhesive film, Euonymus japonicus can be directly transported to the planting site with consolidated soil balls, as shown in Figure 7d. After transplanting Euonymus japonicus, it was found that it grew well. It can be seen that the KGM/CA/PVA ternary blend adhesive also has a good consolidation effect on sandy soil balls, and has no adverse effect on the growth of seedlings.

## 4. Discussion

Because the molecular structure of chitosan contains active groups such as amino, hydroxyl, acetylamino, and electronic pyran ring, it can undergo chemical reactions such as hydrolysis, acylation, carboxymethylation, condensation, and complexation under specific conditions, and can produce derivatives with various physical and chemical functions. Therefore, CA plays an important role in KGM/CA/PVA ternary blend adhesive. However, due to its high value, in order to improve the fluidity of the adhesive and reduce the preparation cost of the adhesive, it is necessary to study the effect of CA content on the consolidation properties of the KGM/CA/PVA ternary blend adhesive. During the study, it was found that the viscosity of the adhesive increased with the increase of CA content at different temperatures. Especially at 60 °C, the viscosity of the adhesive reached a maximum. Therefore, it is necessary to screen the appropriate CA content in combination with the consolidation performance of the consolidated adhesive film on the soil ball.

For the prepared film, it was found that at the same temperature, with the increase of CA content, the film-forming property and toughness of the KGM/CA/PVA ternary blend adhesive were significantly improved. When the temperature was higher than 50 °C, the bending performance of the prepared adhesive film was better as a whole. The SEM micrograph of the adhesive film showed that when the temperature was lower than 50 °C, the links between the cured products on the surface of the adhesive film were weak, the pores formed were large, and there was no cross-linking between the molecular chains, so the adhesive film was brittle. At 70 °C, the surface of the film was densely arranged, and there were many short protrusions on the surface connected to a network structure. It can be seen that the consolidation performance of the film was better when the preparation temperature of the adhesive was higher.

After the KGM/CA/PVA ternary blend adhesive was consolidated on the surface of the soil column, the consolidated film on the surface of the soil column was observed. It was proved again that the film-forming performance of the consolidated adhesive film of the soil column was better with the increase of CA content. Therefore, its bubbles can gradually decrease under the same humidity conditions. When the CA content of the KGM/CA/PVA ternary blend adhesive was the same, the bubbles on the surface of the consolidated film also showed a gradually decreasing trend with the increase of the preparation temperature of the solution. This was because, with the increase in temperature, CA dissolved more thoroughly in the KGM/CA/PVA ternary blend adhesive. CA formed a short cross-linking network structure after blending with KGM and PVA. The film-forming performance of the KGM/CA/PVA ternary blend adhesive was better.

In the test of the compressive strength of the consolidated soil column, it was found that the compressive strength of the consolidated soil column can decrease with the decrease of CA content at the same temperature. At the same time, it was also found that the compressive strength of the consolidated soil column was higher with the increase in adhesive preparation temperature. It was proved again that when the preparation temperature of the adhesive was high, CA was completely dissolved in the adhesive, and blended with KGM and PVA to form a temporary cross-linked network structure, which had a good protective effect on the consolidated soil column. In the study of anti-transport oscillation of consolidated soil column, it was found that compared with the soil column not consolidated with adhesive film, the surface wear of the consolidated soil column was less when the surface of the soil column was protected by adhesive film.

During the exploration of seedling transplanting application, it was found that the KGM/CA/PVA ternary blend adhesive had a good consolidation effect on loam and sandy soil. With the increase of soil ball diameter and the change of soil texture, the requirements for the properties of ternary blend adhesive were slightly different. Therefore, KGM/CA/PVA ternary blend adhesive with different consolidation properties can be prepared according to the actual needs of seedling transplantation.

## 5. Conclusions

This work was performed primarily to study the influence of the preparation conditions of KGM/CA/PVA ternary blend adhesive on the consolidation performance of soil consolidation agents. Its main purpose was to study the effects of the preparation temperature of the glue solution and the content of CA and PVA on the viscosity of the adhesive, the film-forming property of the ternary blend adhesive, the structure of the consolidated film, and its influence on the consolidation performance of the soil column. It was found that the viscosity of the KGM/CA/PVA ternary blend adhesive increased with the increase of temperature and CA content. However, the high viscosity of the adhesive can lead to the decline of the flow performance of the adhesive and then result in the decline of the film-forming performance of the KGM/CA/PVA ternary blend adhesive. Through this study, it was found that the film-forming performance of the adhesive was better when the calcium content was 2–3% and the PVA content was 3–4%. When the preparation temperature was higher than 50 °C, not only the film-forming performance of KGM/CA/PVA adhesive, the brittleness, the moisture, and air permeability of KGM/CA/PVA adhesive film were significantly improved, but also the protection of the adhesive film to the consolidated soil ball (soil column) was enhanced.

In the process of seedling transplanting, it was found that the KGM/CA/PVA ternary blend adhesive had a good consolidation effect on the loam and sandy soil at the root of transplanted seedlings. In the practical application of seedling transplanting, due to the need to consider the preparation cost of the KGM/CA/PVA ternary blend adhesive, while ensuring the soil consolidation effect of the KGM/CA/PVA consolidation agent, it was particularly necessary to consider reducing the content of CA and the preparation temperature of the ternary blend adhesive, so as to achieve the purpose of reducing the preparation energy consumption and cost. Therefore, in the process of practical application, it is necessary to select the polymer soil consolidation agent with the appropriate performance according to the size of the soil ball at the root of seedlings and the texture of the soil.

## Figures and Tables

**Figure 1 polymers-14-02122-f001:**
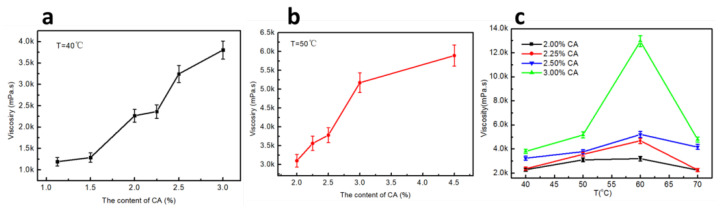
(**a**) Effect of CA content on viscosity of ternary blend adhesive at 40 °C; (**b**) Effect of CA content on viscosity of ternary blend adhesive at 50 °C; (**c**) Effect of CA content on viscosity of ternary blend adhesive at different temperatures.

**Figure 2 polymers-14-02122-f002:**
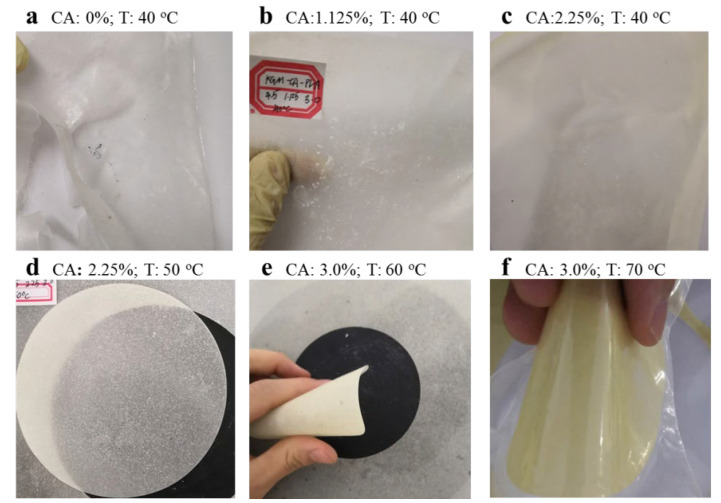
Film-forming properties of different KGM/CA/PVA ternary blend adhesive. Preparation conditions of KGM/CA/PVA ternary blend adhesive: the content of KGM and PVA were 4.5% and 3.0% respectively, the pH value is 5.0.

**Figure 3 polymers-14-02122-f003:**
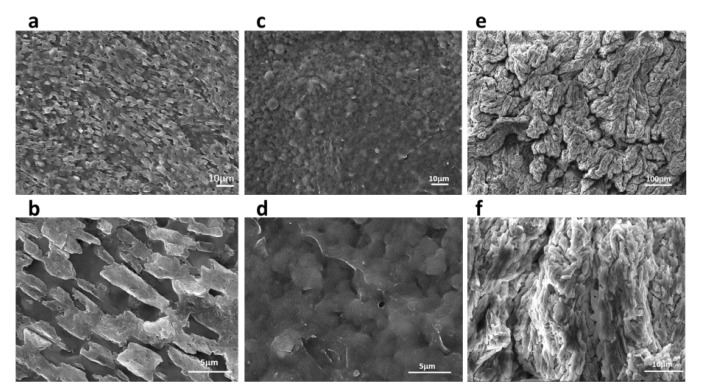
The SEM image of adhesive film. (**a**–**d**) Surface morphology of adhesive film; (**e**,**f**) Cross-section morphology of adhesive film. Preparation conditions of KGM/CA/PVA ternary blend adhesive: the content of KGM and PVA were 4.5% and 3.0% respectively, the pH value is 5.0. (**a**,**b**) the content of the CA was 2.5%, the preparation temperature was 40 °C; (**c**–**f**) the content of the CA was 3%, the preparation temperature was70 °C.

**Figure 4 polymers-14-02122-f004:**
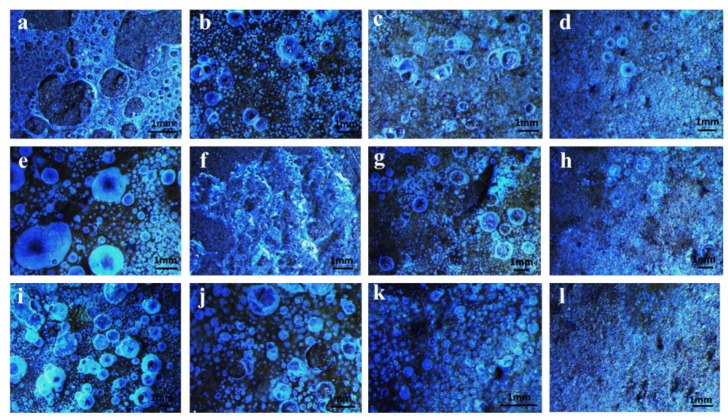
Surface morphology of consolidated adhesive film. (**a**–**d**) Preparation conditions of KGM/CA/PVA ternary blend adhesive: the preparation temperature was 50 °C; The contents of CA were 1%, 2%, 2.5%, 3%, respectively; (**e**–**h**) Preparation conditions of KGM/CA/PVA ternary blend adhesive: the preparation temperature was 60 °C; The contents of CA were 1%, 2%, 2.5%, 3%, respectively; (**i**–**l**) Preparation conditions of KGM/CA/PVA ternary blend adhesive: the preparation temperature was 70 °C; The contents of CA were 1%, 2%, 2.5%, 3%, respectively. Other preparation conditions of (**a**–**l**) KGM, PVA content were 4.5% and 3%, respectively; The pH value of the ternary blend was 5.0; pH of cinnamon soil was 8.5; Particle size of soil was 1 mm.

**Figure 5 polymers-14-02122-f005:**
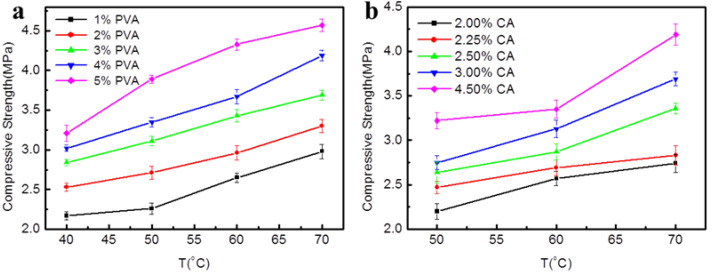
Effect of preparation conditions on compressive properties of consolidated soil columns Preparation conditions of KGM/CA/PVA ternary blend adhesive: the content of KGM was 4.5%, the pH value is 5.0, pH of cinnamon soil was 8.5; Particle size of soil was 1 mm. (**a**) The content of the CA was 4.5%; (**b**) The content of the PVA was 4.0%.

**Figure 6 polymers-14-02122-f006:**
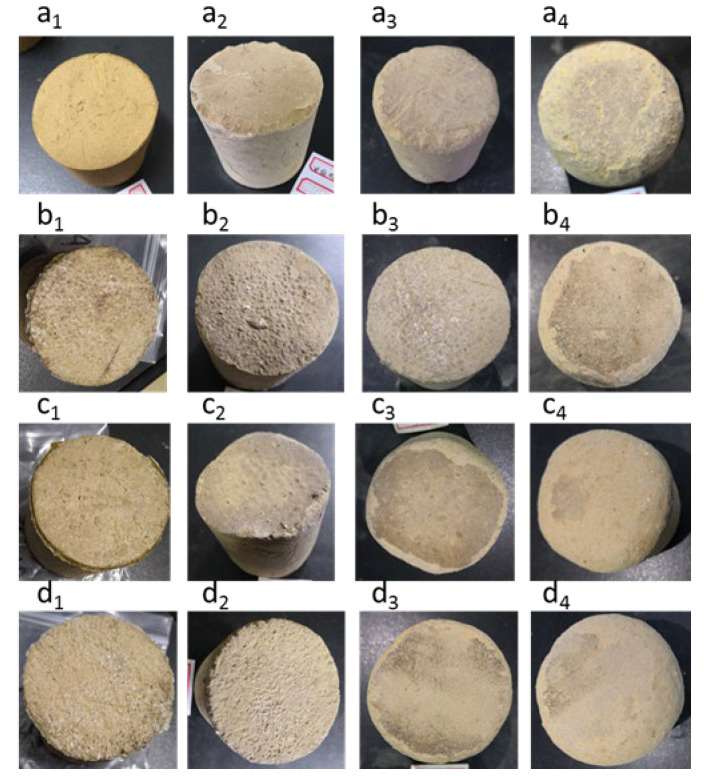
Wear condition of consolidated soil column samples. (**a1**–**a4**) was the blank control group; (**a1**,**b1**,**c1**,**d1**) formed the control group without any oscillation test; The preparation conditions of (**b1**–**b4**) spraying KGM/CA/PVA ternary blend adhesive were 50 °C, the contents of KGM, CA and PVA are 4.5%, 4%, and 4%, respectively, the pH value is 4.5; The preparation conditions of (**c1**–**c4**) spraying KGM/CA/PVA ternary blend adhesive were 60 °C, the contents of KGM, CA and PVA are 4.5%, 4%, and 3%, respectively, the pH value is 4.5; The preparation conditions of (**d1**–**d4**) spraying KGM/CA/PVA ternary blend adhesive were 70 °C, the contents of KGM, CA and PVA were 4.5%, 3%, and 4% respectively, the pH value was 4.5.

**Figure 7 polymers-14-02122-f007:**
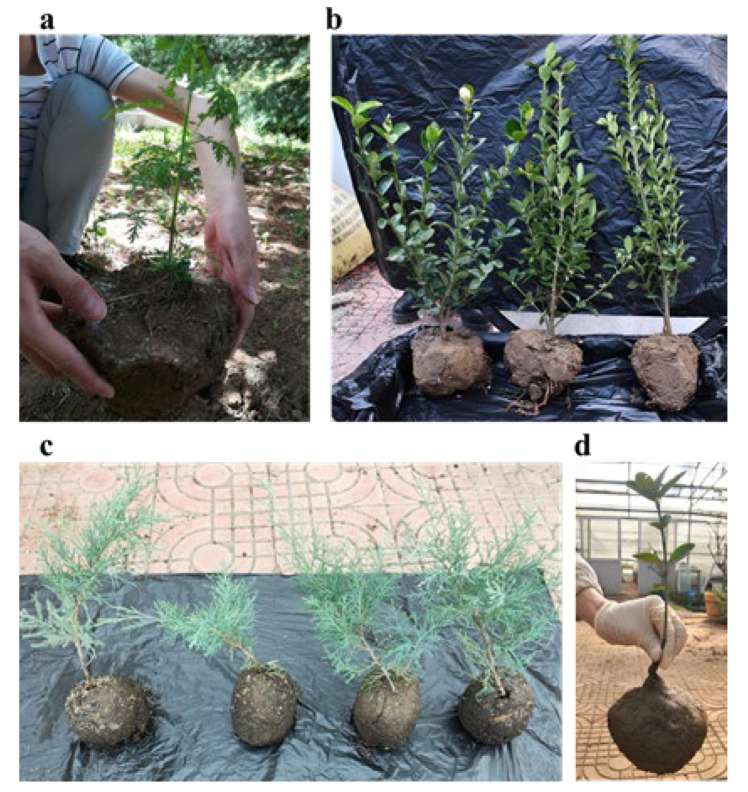
Preliminary application of KGM/CA/PVA ternary blend in seedling transplanting. (**a**) The diameter at breast height (DBH) of sierra salvia was 5 mm; (**b**) The DBH of Euonymus japonicas was 12 mm; (**c**) The DBH of Juniperus Sabina was 5 mm; (**d**) The DBH of Euonymus japonicas was 8 mm.

**Table 1 polymers-14-02122-t001:** ISTA/ASTM Standard Test Method for Simulating Transportation Vibration.

Oscillation Test Sequence	Test Speed (r.min^−^^1^)	Corresponding Frequency (Hz)	Test Time *t* (min)
Test 1	180	3.0	79
Test 2	210	3.5	66
Test 3	240	4.0	60

**Table 2 polymers-14-02122-t002:** Details of Consolidated Soil Column Samples.

Sample No.	Preparation Conditions of Glue Solution
Sample a_1_–a_4_	Blank control
Sample b_1_–b_4_	50 °C, KGM, CA and PVA(4.5%, 4%, 4%), pH4.5
Sample c_1_–c_4_	60 °C, KGM, CA and PVA(4.5%, 4%, 3%), pH 4.5
Sample d_1_–d_4_	70 °C, KGM, CA and PVA(4.5%, 3%, 4%), pH4.5

## Data Availability

This will be made available upon request through the corresponding author.

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
