# Peer review of "Effect of Preparation Conditions on Application Properties of Environment Friendly Polymer Soil Consolidation Agent"

_polymers, 2022, doi:10.3390/polym14102122_

Round 1
Reviewer 1 Report
The manuscript titled “Effect of preparation conditions on application properties of environmental friendly polymer soil consolidation agent” by Wang et al. describes the application of a polymer mixture as a soil consolidation agent. The results are of potential interest, but the Authors need to extensively work on the manuscript to make it suitable for publication. Materials & Methods looks like a transposition of lab protocols and must be rewritten, avoiding useless details (e.g., “adjust the immersion depth of the rotor in the ternary blend adhesive to the corresponding scale mark”, “turn on the switch of the coating machine, “Place the consolidated soil column sample … window of the software for saving”, and similar) and adding relevant information (suppliers of chemicals, chemicals grade, range of concentration of CA, KGM and temperatures tested, details on the instruments like SEM). Grammar and sentence construction must be revised throughout all the manuscript. Conciseness must be privileged: avoid repeating the same concept over and over (e.g., Lines 31-33 in the abstract are not adding information). Results are just summarized in the Discussion section, rather than really discussed. Please find below more detailed comments.
- Check title spelling: substitute “environmental” with “environment”.
- Declare the KGM/CA/PVA acronym in the abstract when first introduced.
- Line 25: higher than what?
- Line 59: check grammar.
- Line 90: “After blending with KGM, the mechanical properties of KGM film can be improved.” Please check this sentence.
- Line 106-107: Do you mean four-neck flask? Why do you need four necks?
- Line 120: which preservative and tackifier?
- Line 130: which amount? Declare the range of solid content tested.
- Why do you sputter gold for such a long time (20-30 min) to prepare the sample for SEM? An extremely thick layer of gold is added to the sample, and this can compromise the investigation of surface morphology.
- Insert table 1 heading.
- Which is the optimal range of viscosity of the glue?
- Line 266: “and there is no cross-linking between the molecular chains”. This cannot be deduced from a SEM micrograph, please remove
- Line 272: what are the requirements on pore size?
- Line 274: “molecular chain” cannot be visualized by SEM, please correct.
- Line 323: correct “PH” to pH.
- Table 2: introduce the table before it appears in the text.
- Line 382: Section heading is not correct.
- The text of section 3.6 can be significantly shortened.
- Figure 1: Add error bars to data points, adjust overlapping of figures.
- Figure 2: CA content and preparation temperature might be added on the corresponding panels instead of listing them in the footnote to facilitate readability.
- Figure 3 and 4: move the text above the figure. Enlarge or add scale bars. KGM and PVA content in the various samples is not clear. Consider adding labels with glue formulation and temperature.
- Figure 5: move the text above the figure.
- Figure 6: check footnote formatting.
- Figure 7: add details on plants.
- Lines 509 and 512: details on funding are contradictory.
Reviewer 2 Report
The “Effect of preparation conditions on application properties of environmental friendly polymer soil consolidation agent” is interesting work. However, I suggested the paper can be improved based on the following comments:
- The authors should provide the summary of results that they received in this work in the abstract section.
- The name of polymer composite in the abstract (first time) should be written as in line 57.
- The novelty of the study is not clear in introduction section. Better to add the outlines sections of work at the end of the introductions.
- In the Preparation of polyvinyl alcohol (PVA) solution, is there a standard for the preparation of the (PVA), if not, what basis where the amounts in this subsection?
- There are differences in the stiles of the subsections font for an example (2.1 and 3.1), please make them in one style based on the MDPI style.
- Figure 1 needs to improve.
- Conclusion so long and need to improve.
Reviewer 3 Report
technical issues: Please check the technical stuff, space, comma ... There is no space line after subtitles in chapter 2. It is difficult to spot subtitles and distinguish them from text .
Put ref in line 22 (In which previous research?).
Line 61 , put ref (In which previous research?).
Line 120, tackifier and preservative can be added . Which one?
Nice work!
Round 2
Reviewer 1 Report
The Authors have addressed all the points of the revision.